



# An open-path observatory for greenhouse gases based on near-infrared Fourier transform spectroscopy

Tobias D. Schmitt[1], Jonas Kuhn[1,a], Ralph Kleinschek[1], Benedikt A. Löw[1], Stefan Schmitt[2], William Cranton[1], Martina Schmidt[1], Sanam N. Vardag[1,3], Frank Hase[4], David W. T. Griffith[5], and André Butz[1,3,6]

[1]Institute of Environmental Physics (IUP), Heidelberg University, Germany
[2]Airyx GmbH, Heidelberg, Germany
[3]Heidelberg Center for the Environment (HCE), Heidelberg University, Germany
[4]Karlsruhe Institute of Technology (KIT), Institute of Meteorology and Climate Research (IMK-ASF), Karlsruhe, Germany
[5]Center for Atmospheric Chemistry, University of Wollongong, Australia
[6]Interdisciplinary Center for Scientific Computing (IWR), Heidelberg University, Germany
[a]now at: Department of Atmospheric and Oceanic Sciences, University of California, Los Angeles, USA

**Correspondence:** Tobias D. Schmitt (tobias.schmitt@uni-heidelberg.de), André Butz (andre.butz@uni-heidelberg.de)

**Abstract.** Monitoring the atmospheric concentrations of the greenhouse gases (GHG) carbon dioxide ($CO_2$) and methane ($CH_4$) is a key ingredient for fostering our understanding of the mechanisms behind the sources and sinks of these gases and for verifying and quantitatively attributing their anthropogenic emissions. Here, we present the instrumental setup and performance evaluation of an open-path GHG observatory in the city of Heidelberg, Germany. The observatory measures path-averaged concentrations of $CO_2$ and $CH_4$ along a $1.55$ km path in the urban boundary layer above the city. We are combining these open-path data with local in-situ measurements to evaluate the representativeness of these observation types on the kilometer-scale. This representativeness is necessary to accurately quantify emissions, since atmospheric models tasked with this job typically operate on kilometer-scale horizontal grids.

For the operational period between Feb. 8 and Jul. 11, 2023, we find a precision of 2.7 ppm (0.58%) and 18 ppb (0.89%) for the dry air mole fractions of $CO_2$ ($xCO_2$) and $CH_4$ ($xCH_4$) in 5-minute measurements, respectively. After calibration, the open-path measurements show excellent agreement with the local in-situ data under atmospheric background conditions. Both datasets show clear signals of traffic $CO_2$ emissions on the diurnal $xCO_2$ cycle. However, there are particular situations, such as under south-easterly wind conditions, where the in-situ and open-path data reveal distinct differences up to 20 ppm in $xCO_2$ most likely related to their different sensitivity to local emission and transport patterns.

Our setup is based on a Bruker IFS 125 HR Fourier transform spectrometer, which offers a spacious and modular design providing ample opportunities for future refinements of the technique with respect to finer spectral resolution and wider spectral coverage to inform on gases such as carbon monoxide and nitrogen dioxide.



# 1 Introduction

Current climate change is driven by anthropogenic emissions of carbon dioxide ($CO_2$) and methane ($CH_4$). The international
community has pledged itself to limit global warming to below 2°C and ideally below 1.5°C compared to the preindustrial era
(UNFCCC, 2015). Thus, substantial reductions of $CO_2$ and $CH_4$ emissions will be required in the coming years. Urban areas
are major contributors to anthropogenic greenhouse gas (GHG) emissions (e.g. Marcotullio et al., 2013) and their reporting
entails considerable uncertainties (Gurney et al., 2021). Monitoring of these emissions through atmospheric concentration
measurements provides a tool to verify reported emission rates and to improve on the precision and spatiotemporal granularity
of emission inventories (e.g. Mueller et al., 2021). This, in turn, might help effective implementation of emission reduction
measures.

Atmospheric $CO_2$ and $CH_4$ concentrations above localized source regions can be measured by spectrometers on satellites
(e.g. Nassar et al., 2017; Kiel et al., 2021), by ground-based sun-viewing spectrometers (e.g. Wunch et al., 2011; Frey et al.,
2019), and by in-situ sensors deployed in networks or on moving platforms (e.g. Shusterman et al., 2016; Fiehn et al., 2020). Re-
cently, techniques have emerged that measure the GHG concentrations integrated along long horizontal absorption paths above
urban areas (e.g. Dobler et al., 2013; Waxman et al., 2017; Griffith et al., 2018). This configuration has the particular advantages
that the observed path-integrated gas concentrations are (1) more sensitive to emission patterns than vertical column-average
concentrations accessible through satellites and ground-based sun-viewing techniques and (2) more representative than in-situ
data for typical grid scales of atmospheric transport models that are used to translate the observed concentration gradients into
emission rates. The latter advantage might be particularly important for urban regions with spatially highly structured source
patterns.

Various absorption spectroscopic techniques have been suggested for long open-path measurements of $CO_2$ and $CH_4$. There
are systems that utilize continuous wave lasers at a few discrete spectral points (Dobler et al., 2013; Lian et al., 2019). Typically
one spectral point is located on a strong absorption feature of the target gas and another one in the transparent region right next
to it. This approach does not provide any information on the shape of the target absorption feature. Tunable diode lasers (TDL)
extend this technique to measurements of a full absorption line, with the potential to measure a few spectrally close absorption
lines (Bailey et al., 2017). Laser frequency combs, e.g. operated in a dual-comb spectroscopy setup (Coddington et al., 2016),
can improve on this by covering a few hundred wavenumbers and measuring full rotational-vibrational absorption bands
with dozens of lines (Truong et al., 2016; Waxman et al., 2019). This can provide access to temperature information within
the spectrum. Fourier transform spectroscopy (FTS) can measure across spectral intervals of several thousand wavenumbers
simultaneously giving access to a plethora of species at the same time. FTS open-path systems working in the mid infrared
(MIR) are well established (Wiacek et al., 2018; Bai et al., 2020; You et al., 2021). MIR FTS systems are typically limited by
low brightness of the light source resulting in maximum atmospheric absorption paths lengths of a few hundred meters. Near
infrared (NIR) FTS systems can make use of hotter thermal light sources like halogen lamps and with that achieve path lengths
exceeding kilometers.





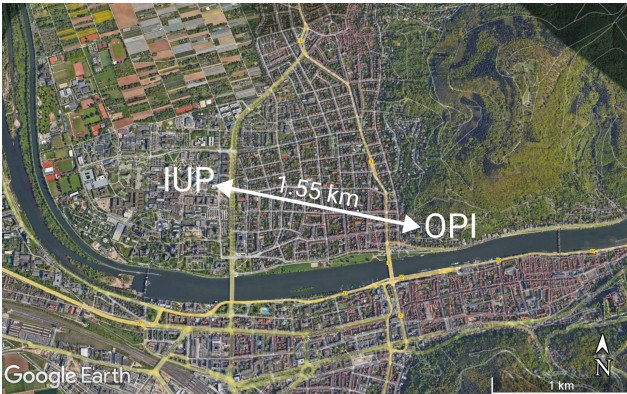

**Figure 1.** Aerial view of the measurement path above the city of Heidelberg, Germany. All active components are located at the western end of the path at the Institute of Environmental Physics (IUP). The reflector array is located on the roof of the Old Physics Institute (OPI).

Such NIR FTS open-path measurements for $CO_2$ and $CH_4$ were pioneered by Griffith et al. (2018) in a demonstration study for the city of Heidelberg, Germany. Their system was built around an IRCube FTS manufactured by Bruker Optics - a compact, robust spectrometer. Since then, their setup was improved and developed into a portable version, suitable for field deployments (Deutscher et al., 2021). We followed a complementary path and built a new open-path system in Heidelberg with the goal of designing a permanent atmospheric observatory suitable for future methodological explorations and for long-term measurements of $CO_2$ and $CH_4$ above Heidelberg. To this end, we used a Bruker Optics IFS 125HR FTS that, due to its modular and spacious layout, allows for configuring the spectral resolution and for deploying various detectors giving access to various spectral ranges.

Here, we report on the initial setup of the observatory and its performance for measuring $CO_2$ and $CH_4$ concentrations along a 1.55 km path above Heidelberg, essentially mimicking the overall configuration of the demonstrator study by Griffith et al. (2018) but demonstrating the performance of the IFS 125HR FTS. Section 2 elaborates on the experiment and the methods used, with a particular focus on how the 125HR FTS is used in the long open-path setup. Section 3 reports on the performance for months-long $CO_2$ and $CH_4$ observations and discusses the observations in comparison to the local in-situ records. Section 4 concludes the study with a discussion of future exploration possibilities.

## 2 Experiment and methods

### 2.1 Deployment configuration

The location of the measurement path above the city of Heidelberg is shown in Figure 1, which is the same as in Griffith et al. (2018). All active components (spectrometer, light source) are located at the west end of the path in the roof-top observatory (Figure 2, left) of the Institute of Environmental Physics (IUP). A reflector array is located at the east end of the path on the roof of the Old Physics Institute (OPI) (Figure 2, right). The length of the path is $1553\,\mathrm{m} \pm 1\mathrm{m}$ as determined with a laser





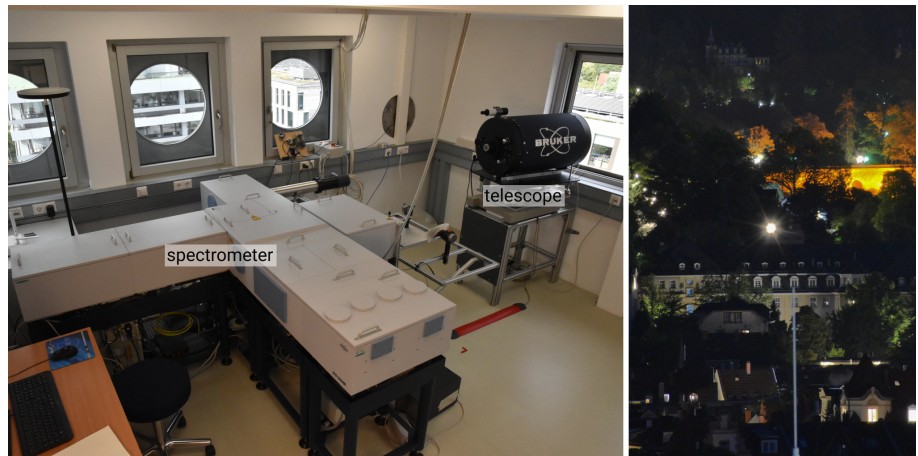

**Figure 2.** Left: Overview of the roof-top observatory of the IUP with the IFS 125HR spectrometer in the center, as well as the telescope pointing out a window. Right: View along the light path at night, with the reflector array on top of the OPI shining bright in the center of the picture.

rangefinder. The air inlet for the in-situ analyzer, which is used for comparison, is located at the western end, on the roof of the IUP. The in-situ analyzer is a cavity ring-down spectroscopy (CRDS) system (G2201-i, Picarro, Inc., Santa Clara, CA). The whole system including drying and calibration setup is described in detail by Hoheisel et al. (2019) and Hoheisel (2021). Measurements of temperature, wind speed, wind direction, humidity and pressure are regularly taken by a weather station, located on the roof of the IUP.

The setup in the current hardware configuration is operating since Feb. 8, 2023 and the measurements discussed here extend until Jul. 11, 2023. During this time, the instrument was operated in different configurations concerning the integration time as well as spectral resolution. Further, the measurements were interrupted at several occasions, when the FTIR instrument was needed for other experiments.

## 2.2 FTIR setup and open-path optics

Figure 3 shows a sketch of the optical setup. The internal light source of the spectrometer (a 20 W halogen lamp) is collimated and transmitted through the interferometer of the IFS 125HR (focal length of collimator: 418 mm, field stop: 2 mm). The modulated beam of light is coupled into a bundle of fibers ($6 \times 200\,\mu$m, length 5 m, VIS-IR silica), which we call the transmitting fibers. At the other end, the transmitting fibers are grouped around a single, central fiber and are positioned close to the focus of the telescope mirror (focal length: 813 mm, diameter: 406 mm). The telescope mirror collimates the modulated light of the transmitting fibers and sends it through the atmosphere to an array of retro-reflectors (array consists of 41 solid UV quartz cube-corners, each with a diameter of 53 mm). The retro-reflector array returns the beam of light back to the telescope, where it is coupled into the single, central fiber (200 µm, length 5 m). At the other end of this receiving fiber the light is focused on the photodiode of an external detector of the spectrometer. This type of open-path telescope design and fiber system was





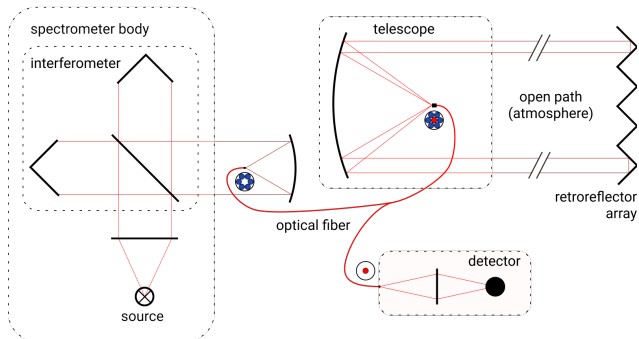

**Figure 3.** Schematic drawing of the optical setup. The light source is modulated in the interferometer of the spectrometer and coupled into a bundle of six fibers (blue). These six fibers transport the light to a telescope, where it is collimated and sent along the open-path on an array of retro reflectors. The reflectors return the beam of light to the telescope where it is collected by a single, central receiving fiber (red). From this fiber the light is focused on a detector.

originally developed for DOAS measurements. An extensive description of the system and an in-depth analysis of the optical throughput is provided in Merten et al. (2011).

For the target gases $CO_2$ and $CH_4$ relevant here, the detector is an InGaAs diode with a lower frequency cut-off around $4200\,\mathrm{cm}^{-1}$. The quartz cube corners also cut off at around $4200\,\mathrm{cm}^{-1}$. To high frequencies, the spectrum is limited by a filter at around $9200\,\mathrm{cm}^{-1}$. The current setup also disposes of Si and GaP detectors for the UV/visible spectral range, but these were

not used here.

Compared to Griffith et al. (2018), the new setup modulates the light in the interferometer before sending, i.e. it is the interferogram (instead of the lamp spectrum) that travels along the absorption path. This has the advantage that scattered sunlight does not disturb the spectrum and thus, background measurements (with the lamp blocked) are not required. Any light that is collected in the telescope but not modulated by the interferometer contributes to the total photon-flux on the detector

only increasing shot-noise, but not confounding the spectral information.

The FTS instrument itself was operated in three different configurations throughout the reported timeframe. Our default mode of operation was chosen similar to Griffith et al. (2018) to allow for easy comparison. In this mode, we average $34$ interferograms over a time span of approximately $5\,\mathrm{min}$. The interferograms are measured double sided, using maximum optical path difference ($\mathrm{OPD_{max}}$) of $1.8\,\mathrm{cm}$, which corresponds to a nominal resolution of about $0.55\,\mathrm{cm}^{-1}$. The scanner

operates with $10\,\mathrm{kHz}$ scanning speed at the reference HeNe laser line. The Fourier transform is performed using the standard Merz phase correction and a Norton-Beer Medium apodization function. These settings were employed from Feb. 8 until Mar. 22 and from May 25 until Jul. 11. A typical spectrum recorded with these settings is shown in Figure 4.

Apart from these default settings, we experimented with shorter averaging times of approximately $1\,\mathrm{min}$ from Mar. 23 until May 10, as well as with higher resolutions of approximately $0.11\,\mathrm{cm}^{-1}$ ($\mathrm{OPD_{max}} = 9\,\mathrm{cm}$) from May 11 until May 24. Table 1



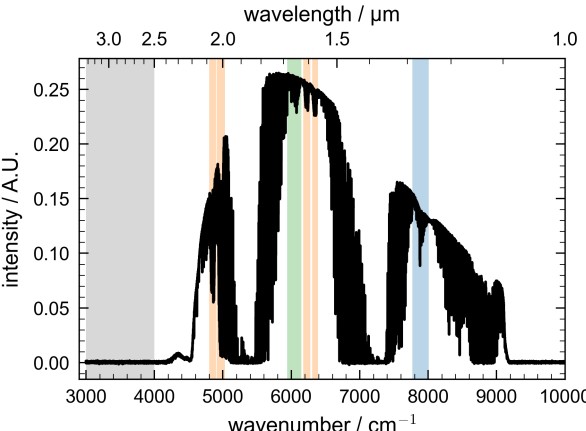

**Figure 4.** Typical spectrum measured over $5\,\mathrm{min}$ with a resolution of $0.55\,\mathrm{cm}^{-1}$ and a SNR of 600. Target windows are highlighted in color: $O_2$ (blue), $CO_2$ (orange), $CH_4$ (green). The off-band window, where we gather noise information is marked in grey.

**Table 1.** Summary of the settings for the FTIR instrument.

| setting ID | $OPD_{max}$ | nominal resolution | coadded scans | measurement duration | repetition period | typical SNR | interferogram type |
|:---:|:---:|:---:|:---:|:---:|:---:|:---:|:---:|
| 1 | $1.8\,\mathrm{cm}$ | $0.55\,\mathrm{cm}^{-1}$ | 34 | 306.7s | 311.4s | 690 | double sided |
| 2 | $1.8\,\mathrm{cm}$ | $0.55\,\mathrm{cm}^{-1}$ | 6 | 52.7s | 57.4s | 240 | double sided |
| 3 | $9.0\,\mathrm{cm}$ | $0.11\,\mathrm{cm}^{-1}$ | 16 | 285.6s | 295.3s | 130 | single sided |

lists the details on the utilized instrument settings and the resulting times. All measurements are resampled to a regular $5\,\mathrm{min}$ grid to combine the different modes of operation.

Signal-to-noise-ratio (SNR) was determined by taking noise information from the white noise in the off-band and using the spectral intensity at around $6300\,\mathrm{cm}^{-1}$ as a measurement for signal to allow for comparison with Griffith et al. (2018). SNR was typically between 500 and 700 for the default mode of operation.

## 2.3 Spectral analysis and trace gas retrieval

We retrieve the path-integrated column densities (CDs) of the target gases from the measured spectra with a spectral fit. The forward model simulates the spectra based on the following input parameters: assumed gas column density (CD), spectral absorption cross-sections, temperature, and pressure. It further takes into account the instrument line shape (ILS) of the FTS, a spectral shift, and a polynomial to represent the continuum. The radiative transfer in the forward model is limited to a simple implementation of Beer-Lambert's law. The forward model and retrieval are implemented in python.



**Table 2.** Spectral windows and corresponding fitted trace gases. The main absorbers are highlighted in bold for each window.

| window ID | spectral range / $cm^{-1}$ | fitted species |
|:---:|:---:|:---|
| 1 | 4800–4900 | $\textbf{CO}_\textbf{2}$, $\textbf{H}_\textbf{2}\textbf{O}$, **HDO** |
| 2 | 4910–5030 | $\textbf{CO}_\textbf{2}$, $\textbf{H}_\textbf{2}\textbf{O}$, **HDO** |
| 3 | 5940–6145 | $\textbf{CH}_\textbf{4}$, $CO_2$, $H_2O$, HDO |
| 4 | 6173–6276 | $\textbf{CO}_\textbf{2}$, $H_2O$, HDO |
| 5 | 6308–6390 | $\textbf{CO}_\textbf{2}$, $H_2O$, HDO |
| 6 | 7770–8010 | $\textbf{O}_\textbf{2}$, $O_2$(CIA), $H_2O$ |

The absorption cross-sections are calculated using the Voigt implementation of the HITRAN interface HAPI (Kochanov et al., 2016). We use line-by-line data provided in the HITRAN2020 database (Gordon et al., 2022). For the $CO_2$ bands at $5000\,cm^{-1}$, we use additional line mixing corrections (Hartmann et al., 2009; Lamouroux et al., 2010). In the $O_2$ band at $7900\,cm^{-1}$, we account for collision induced absorption (CIA) by fitting a pseudo-absorber. The absorption cross-sections for

this pseudo-absorber are calculated using a temperature and pressure dependent air mass (via the ideal gas law) and the CIA data currently provided by HITRAN for $O_2$–Air interaction (Karman et al., 2019).

For fitting, the spectra are cut into various spectral windows, which contain the spectral signal of several target gases. Table 2 lists the windows used here, their spectral range and all fitted species. All species are assumed with the HITRAN2020 standard mixture of isotopologues, except for $H_2O$ and HDO, which we fit separately.

The retrieval uses a Levenberg-Marquardt implementation for least-squares optimization. The differences between forward simulated spectra and the measurements are weighted by a noise estimation taken from the off-band ($3000\,cm^{-1} - 4000\,cm^{-1}$). We optimize the trace gas CD for each species. If gases need to be considered for several spectral windows, we fit their CDs separately per window and *a posteriori* calculate an average CD weighted by the propagated noise error. Temperature is fitted, while pressure is kept fixed at the measurement of our weather station. A spectral shift, as well as a background polynomial of

degree 2 is also fitted in each window. In window 6, this polynomial is of degree 5, due to the width of the window.

The ILS of the instrument is described by a fixed set of parameters according to the simple model described in Hase et al. (1999). The parameters are the field of view (FOV) of the instrument and a single parameter for modulation efficiency and phase error each. We estimate modulation efficiency and phase error using 200 consecutive measurements taken on Feb. 26, 2023. For each measurement a retrieval is performed as described above, but with ILS modulation efficiency and phase error

as additional free parameters. The mean of these results for modulation efficiency and phase error is taken to simulate the ILS. The FOV is calculated to $4.78\,\mathrm{mrad}$ from geometrical parameters of the instrument.

## 2.4 Dry air mole fractions

For comparison to external $CO_2$ and $CH_4$ data, e.g. provided by in-situ measurements or by models, it is convenient to translate the path-average CDs into path-average dry air mole fractions (e.g. in units ppm). In the scope of this study we calculate dry





air mole fractions $x_{dry}$ by ratioing the measured CD of the target gas $CD_{target}$ with the measured CD of $O_2$ $CD_{O_2}$ and multiplying with the $O_2$ dry air mole fraction:

$$x_{dry} = \frac{CD_{target}}{CD_{O_2}} \cdot 0.20946 \qquad (1)$$

It is also possible to forgo the measured $O_2$ CD and calculate the dry air mass from information on temperature, pressure, humidity and pathlength. Since these values can be measured quite precisely, the results are typically up to an order of magni-

tude more precise than the measured $O_2$ CD, so that the precision of dry air mole fractions should only be dominated by the precision of the target gas CDs. Because of this, Griffith et al. (2018) employed this method.

To compare the two methods, we calculated CDs of dry air $CD_{dry}$ as follows: For temperature we use the retrieved path-averaged temperature $T$ (as described in subsection 2.3), for pressure the meteorological pressure $p$ from the weather station at the institute and the pathlength $l$ as measured with a laser rangefinder. As a measure of humidity, we decided to directly

take the measured CD of $H_2O$ vapor $CD_{H_2O}$ and subtract it from the calculated wet air CD. This leaves us with the following equations (with $k_B$ being the Boltzmann constant):

$$x'_{dry} = \frac{CD_{target}}{CD_{dry}}, \qquad CD_{dry} = \frac{p}{k_B T} l - CD_{H_2O} \qquad (2)$$

## 2.5   Precision of time series

To estimate the precision of measured quantities (like target gas $CD_{target}$, dry mole fractions $x_{dry}$, or temperature.), we took

an empirical approach: We calculated the standard deviation among all differences between two consecutive measurements. By dividing the standard deviation by $\sqrt{2}$, we get the standard deviation for a single measurement. This is our measure for the mean precision of time series of measured quantities.

## 2.6   Calibration to in-situ data

The spectroscopically inferred open-path dry mole fractions require calibration to ensure compatibility with other data sources

such as in-situ concentration measurements. This is necessary since the spectroscopic parameters, such as line strengths and broadening coefficients, tabulated in HITRAN and feeding into the calculation of absorption cross-sections, typically show inconsistencies between absorption bands and their calibration has errors (e.g. Birk et al., 2021). In-situ instruments are calibrated to known standard gas mixtures and hence they are highly accurate (Hall et al., 2021). We operate such in-situ measurements at the IUP, hence we can calibrate our open-path data to the in-situ data. However, in-situ and open-path instruments, by nature

and intention, measure different bodies of air. Thus, the calibration should be performed under atmospheric conditions when the local mole fractions at IUP are representative of the path-averaged mole fractions across the city i.e. when there is no relevant effect from local emissions compared to the bulk concentration in the boundary layer. To select such conditions, we examine the histograms of $xCO_2$ and $xCH_4$ measured by the two instruments. Figure 5 shows that theses histograms have a peak at low mole fractions and a long tail toward high mole fractions. Since our sampling area and period is dominated by emissions rather

than uptake for both species the tail corresponds to episodes where local emissions influence the concentration measurements



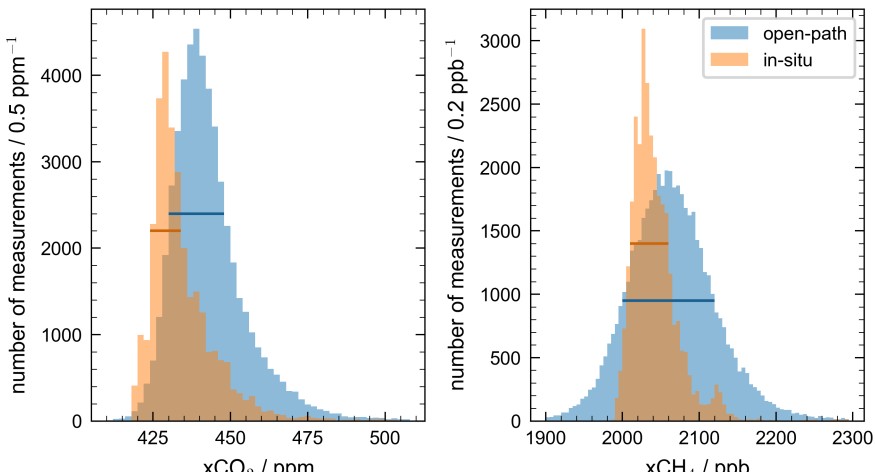

**Figure 5.** Histogram of measured dry air mole fractions for $xCO_2$ (left) and $xCH_4$ (right) from the in-situ and open-path instrument. The expected peak of low concentration when measuring background air is clearly visible for both instruments. For $xCO_2$ ($xCH_4$), there are 33426 (37633) measurements in the selected peak area (blue and orange bar respectively) for the open-path instrument and 17064 (22127) measurements for the in-situ analyzer. 13633 (16525) of those are coincident in time. Calculating calibration factors for all these measurements and taking the mean yields a calibration factor of 0.9784 (0.9887).

significantly and thus, horizontal gradients are likely. The peak at lower concentrations corresponds to background conditions for which local emissions are less important and thus, we expect smaller gradients. To derive a calibration factor, we visually identify these peaks and select the background measurements by filtering for observations which are coincident in both peaks. The calibration factor is the mean of the ratio between the open-path and in-situ data for all these background measurements. It amounts to approximately 0.9784 for $xCO_2$ and 0.9887 for $xCH_4$, respectively.

# 3 Results

## 3.1 Quality of spectra and fit

Residuals of the spectral fit are illustrated in Figure 6 for the spectral windows (see Table 2). Spectral residuals of individual 5 min measurements are typical below $1\%$ of the spectral intensity and mostly dominated by noise, while averaging over longer timescales reveals systematic patterns. In windows 1 and 2, the systematic effects are on the same order of magnitude as the noise while in the other windows they are mostly smaller than the noise.

Fitting the path-averaged temperature instead of imposing it from the weather station data yields a noticeable improvement in fit quality and results in more realistic temperature diurnal cycles. This is especially true in the early morning hours with substantial temperature gradients along the path. Most temperature information comes from the $2\,\mu m$ windows (windows




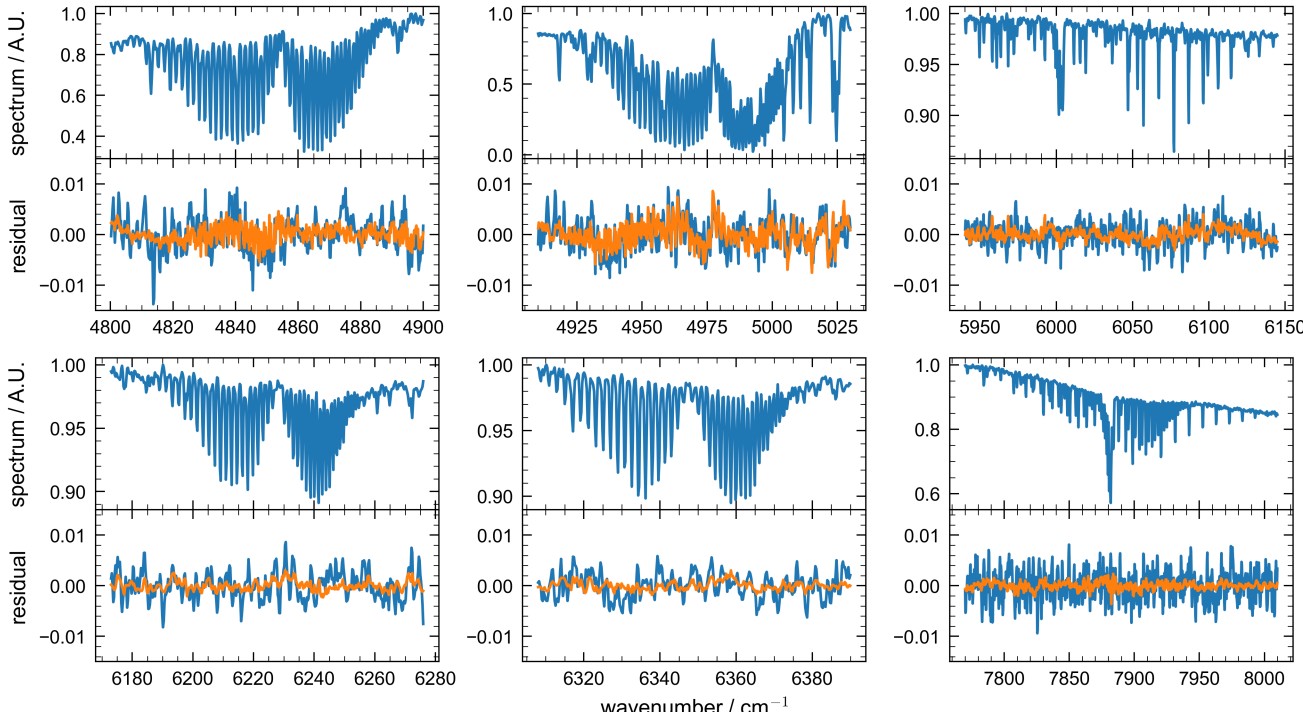

**Figure 6.** Typical spectra (upper subpanels) and fit residuals (lower subpanels, measured minus simulated) for 5 min averages (blue) over-layed with residuals averaged over the whole measurement period (orange) for the 6 retrieval windows (panels) defined in Table 2. Spectra are scaled to the maximum intensity for each window respectively.

1 and 2). The estimated path-averaged temperature is on average $0.7\,\mathrm{K}$ lower than the temperature at the weather station. We find that sensitivities studies under the neglect of line-mixing in the $2\,\mu\mathrm{m}$ windows (windows 1 and 2) resulted in path-averaged temperature estimates that were low-biased by $2\,\mathrm{K}$ to $6\,\mathrm{K}$ with respect to the weather station records. This bias is consistent with the findings of Griffith et al. (2018) who neglected line-mixing and found similar low-biased path-averaged temperature information. Thus, consideration of spectroscopic line-mixing effects is required to accurately represent the actual path-averaged temperature and to improve on fit quality.

### 3.2 Precision of target gas CDs and dry air mole fractions

Table 3 lists the relative precision for $CO_2$ and $CH_4$ CDs and their dry air mole fractions $xCO_2$ ($2.5\,\mathrm{ppm}$, $0.58\,\%$) and $xCH_4$ ($18\,\mathrm{ppb}$, $0.89\,\%$) inferred by ratioing the target gas CDs with the measured $O_2$ CDs. The precisions for dry air mole fractions are comparable to those achieved by Griffith et al. (2018) ($1.7\,\mathrm{ppm}$ for $xCO_2$ and $23\,\mathrm{ppb}$ for $xCH_4$), while they are substantially less precise than Deutscher et al. (2021) ($0.28\,\mathrm{ppm}$ for $xCO_2$ and $2.1\,\mathrm{ppb}$ for $xCH_4$; in 3 min measurements). Deutscher et al. (2021) mostly achieved this by a fixed connection of FTS instrument and telescope, replacing the fiber coupling





**Table 3.** Relative precisions of measured total CDs and dry air mole fractions for retrieved species.

| species | CDs / % | dry mixing ratios / % |
|---------|---------|----------------------|
| $CO_2$  | 0.70    | 0.58                 |
| $CH_4$  | 0.92    | 0.89                 |
| $O_2$   | 0.77    | –                    |
| $H_2O$  | 1.08    | –                    |

with a mirror and beam-splitter setup that allows for significantly higher optical throughput by more than an order of magnitude, substantially improving the spectral SNR and hence the precision of the retrieved trace gases. Due to size and weight, this type of configuration is not possible when using the IFS 125 HR FTS. Also, we do only find minor changes in precision for xCO$_2$ and xCH$_4$ for the different settings listed in Table 1, even though the measurements differ significantly in spectral SNR due to different temporal and spectral resolution. The $1\,\mathrm{min}$ measurements show the same precisions for target gases once averaged on the same $5\,\mathrm{min}$ timescale, showing that the precise is not dominated by real fluctuations during the time of a single measurement. The high resolution measurements show the same target gas precisions as the low resolution counterparts. Since the line-depth is mostly dominated by the ILS for the low resolution parameters, it increases with higher resolution and compensates for the worse spectral SNR.

Interestingly, our dry air mole fractions show a slightly higher level of precision than the respective CDs. We conclude that there is a systematic effect, which affects all retrieved CDs by a common similar factor. This systematic effect then at least partially cancels when dividing the target gas CDs by the measured $O_2$ CDs. A candidate for causing such an effect could be changes of the refractive index along the light path during the recording of the interferograms. These changes induce slight variations of the transmittance of the optical system with time and effectively function as a modification to the apodization function, slightly altering the ILS. Hence, calculating the dry air mole fractions by dividing the target gas CDs by the $O_2$ CDs retrieved from the same measurement, partially compensates for this effect.

As expected, the dry air masses calculated via Equation 2 way show a significantly higher precision then using measured $O_2$ CDs ($0.089\,\%$ instead of $0.77\,\%$). But this reverses if we look at the precision of the dry air mole fractions calculated from these air masses: For example for $CO_2$, where the total CD is measured to a precision of $0.70\,\%$, the dry mixing ratios calculated via Equation 1 show a precision of $0.58\,\%$, while calculated via Equation 2 yields $0.73\,\%$. So when dividing with the uncorrelated, but highly precise calculated total air masses, the noise propagates as expected and results in less precise dry air mole fractions. This strengthens the argument, that both the $CO_2$ (or $CH_4$) CD and the $O_2$ CD are effected by a systematic noise, which at least partially cancels out when dividing one by the other.

## 3.3 Timeseries of CO$_2$ and CH$_4$

Figure 7 shows the timeseries of the dry air mole fractions xCO$_2$ and xCH$_4$ after calibration and after filtering for low signal measurements (for example due to fog or dense rain). Along the time period from Feb. 6 until Jul. 11 we achieve a data coverage



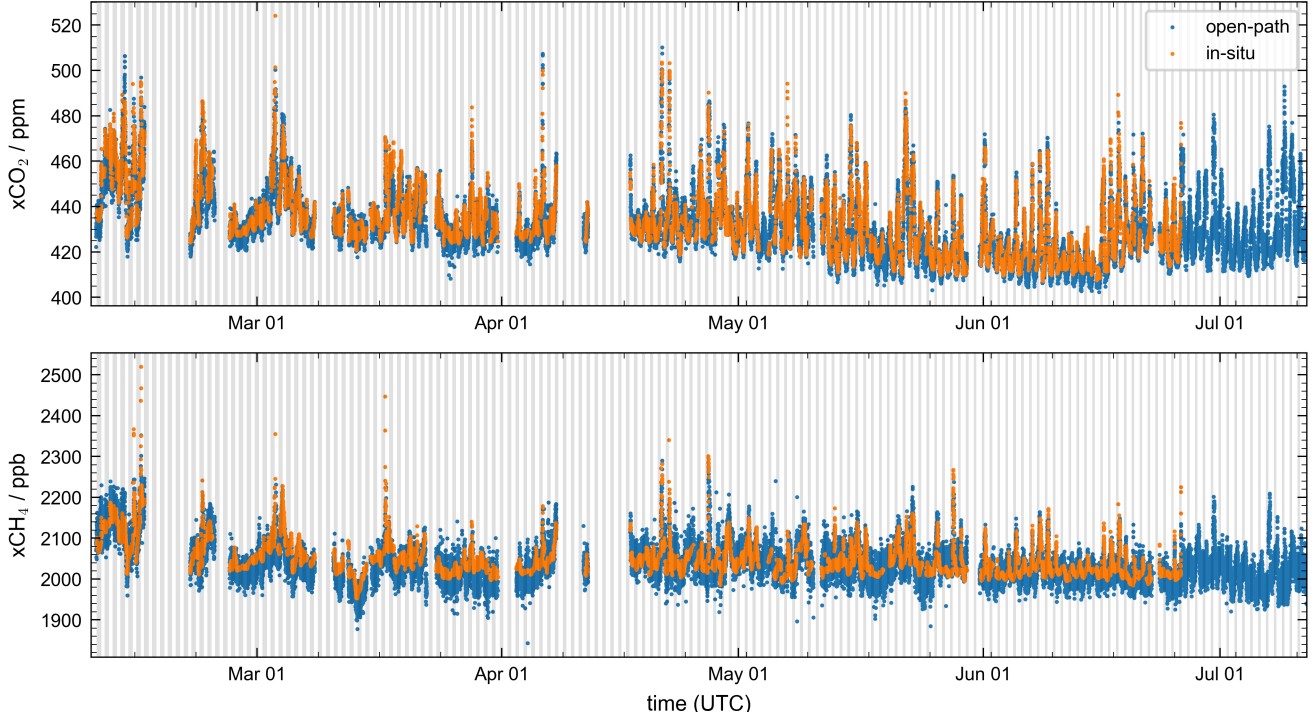

**Figure 7.** Comparison between open-path (blue) and in-situ (orange) measurements, for dry mole fractions of $CO_2$ (upper panel) and $CH_4$ (lower panel). The shown open-path data covers the period from Feb. 6 until Jul. 11, including gaps of a few days. In-situ measurements were available until Jun. 26.

of 78 % for 10 minute intervals. Months where the instrument could be fully dedicated to the open-path measurements achieve even better coverage of up to 92 % in June. We find good agreement between open-path and in-situ data. Both records reveal a

decrease of the background $xCO_2$, especially from mid of April onwards, as expected from enhanced biospheric activity in the Northern hemisphere in spring and summer. On top of this large scale trend, the signal is dominated by the diurnal cycle with regular night time enhancements and by events related to regional and local emissions and transport patterns. The two records also agree well for $xCH_4$, but the timeseries show less regular diurnal cycle structures. For both gases, the open-path data show a greater scatter than the in-situ data as expected from the precision estimates.

Figure 8 shows the statistics of the differences between the open-path and in-situ records of $xCO_2$ and $xCH_4$. While the majority of the differences center around zero with a spread that corresponds to the precision of the records, the statistics shows a significant tail for $xCO_2$, with the in-situ data being larger than the open-path data.

While the records largely agree, Figure 9 showcases events for $xCO_2$, and to an extend for $xCH_4$, where the open-path and in-situ data differ. In the period Mar. 17 to Mar. 19 in-situ and open-path data align during the day but they differ by up to

20 ppm in $xCO_2$ during the night, while no noticeable difference is present in the $xCH_4$ records. Both of these nights have in



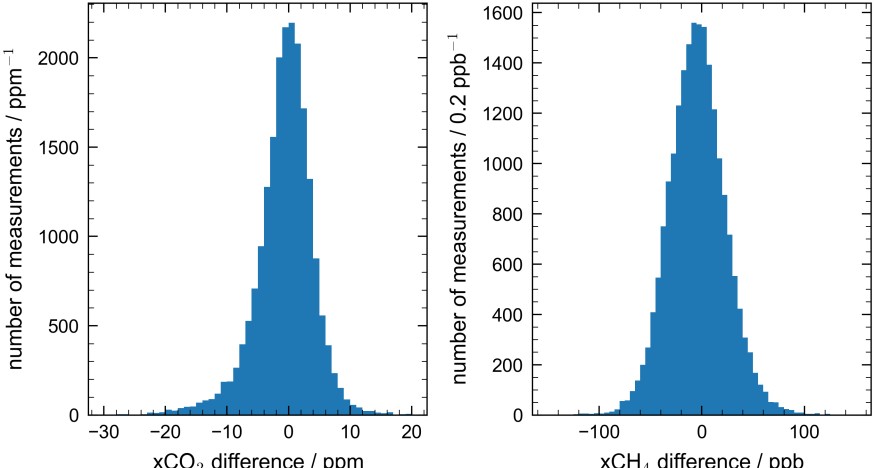

**Figure 8.** Difference between the two instruments (open-path minus in-situ) for $xCO_2$ (left) and $xCH_4$ (right). A clear central peak around 0 is visible for both gases. For $xCO_2$, measurements where both instruments significantly deviate from each other tend to bias towards lower values, so lower concentrations measured by the open-path system and higher concentrations measured by the in-situ system. For $xCH_4$, there is no significant bias.

common, that the predominant wind direction (as measured at the institute) is from South-East (135°). Figure 10 shows the differences between open-path and in-situ plotted over wind-direction. The distribution indeed shows a systematic difference in $xCO_2$ for wind directions around 135° with a gentle decline of the differences towards winds from the east and a relatively sharp transition towards winds from the south. The city center of Heidelberg at the mouth of river Neckar valley is located at

around 135°. One hypothesis is, that in this wind situation the residential area below the open-path is flushed with $CO_2$-poor air from the river valley due to a deflection of air masses when exiting the Neckar valley, while the in-situ station still samples the the $CO_2$-rich urban air masses. This invites further research on whether high-resolution meteorological models can represent such effects and what the representativeness of the open-path and in-situ records are.

For $xCH_4$, the picture is less clear. For most wind directions, the in-situ instrument measures slightly higher $xCH_4$ con-

centrations than the open-path instrument. The most notable feature is a peak of relatively higher open-path concentrations for winds from the East and Northeast. All of this, however, is barely significant. In time series, differences in $xCH_4$ typically show in form of sharp emission spikes which are covered by both instruments to a different extend, most likely due to local source location. The largest sources of CH4 in the area are leakages from the natural gas distribution system and emissions from the sewer system. It seems reasonable, that the open-path system with its larger area footprint would see such emissions on a more

regular basis than the in-situ instrument and if the in-situ instrument is located within such an emission plume, that the total enhancement is more diluted in the total CD measured by the open-path system. A detailed assessment of such differences, however, requires local modelling of airmass transport.



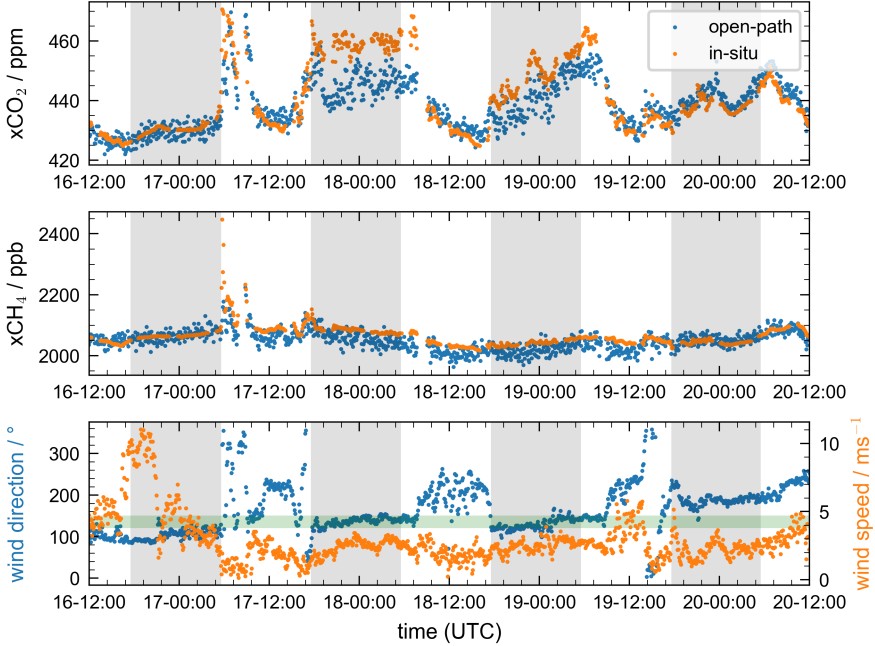

**Figure 9.** Events of interest from Mar. 16 to Mar. 20, 2023. On Mar. 17, shortly after sunrise, $xCH_4$ (middle panel) increases by several 100 ppb in a sharp spike for the in-situ measurement (orange), but only by a fraction of that for the open-path (blue). Further, during the two subsequent nights the in-situ records show a significant enhancement over the open-path instrument for the measured $xCO_2$ (top panel). For both of these latter events, the wind direction (lower panel, blue) is quite steady around $135° \pm 15°$ (green horizontal band). The wind speed (lower panel, orange) does not show any remarkable features during this time.

Figure 11 shows the diurnal cycles for $xCO_2$ for weekends and weekdays averaged over the entire reported period. There is a clear difference between weekends and weekdays for the morning hours, which is related to $CO_2$ exhaust from traffic being emitted into a shallow morning boundary layer during weekdays but not during weekends. The enhancement in the morning amounts to roughly 5 ppm while the afternoon rush-hour results in smaller enhancements since boundary layer mixing processes are more effective in the afternoon than in the morning. Comparing the in-situ and open-path data, the agreement is excellent for the weekends with average differences of less than 1 ppm during the day and slightly larger differences during night. On the weekdays, however, the in-situ data rather systematically indicate 1 to 2 ppm greater $xCO_2$ than the open-path data. This might point to a local signal that contributes more to the in-situ samples which is not represented on the 1.5 km averaging scale of the open-path experiment. While such representativeness issues might appear small, they become important when aiming at quantification and verification of local emissions with atmospheric models.





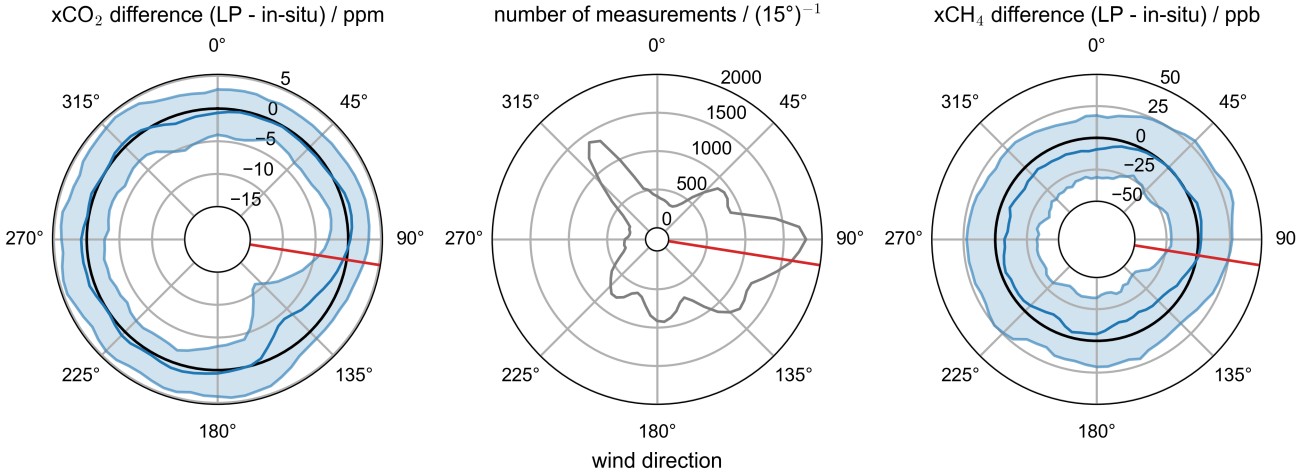

**Figure 10.** Difference between the two instruments (open-path minus in-situ) for xCO$_2$ (left panel) and xCH$_4$ (right panel) depending on wind direction. Plotted differences are averages over all measurements which have a wind directions within a $\pm15°$ arc and a wind speed above $1\,\mathrm{ms}^{-1}$. The blue band gives the one sigma percentile of the distribution. An obvious feature for the wind direction from approximately 135° is visible for xCO$_2$, but not for xCH$_4$. The open-path system measures on average significantly lower values for xCO$_2$ than the in-situ system for wind coming from the south east.

## 4 Conclusions

We report on the successful development, setup, and performance evaluation of an open-path greenhouse gas observatory in the city of Heidelberg, Germany. The system is based on the IFS 125HR FTS manufactured by Bruker Optics, which makes our setup highly versatile regarding spectral resolution, as well as regarding detector options, and hence spectral coverage. However, this choice of instrument does not provide the portability demonstrated by the setup of Deutscher et al. (2021). Our system mainly operates at a standard spectral resolution of $0.55\,\mathrm{cm}^{-1}$, with experiments run at $0.11\,\mathrm{cm}^{-1}$. The spacious and modular design of the FTS enables modulating the light beam coming from a halogen light source first, before the light beam travels across the $1.5\,\mathrm{km}$ path (one-way) above the city. Sending the interferograms instead of the unmodulated light beam along the absorption path enables measurements during day and night without the need to correct for scattered sunlight. Continuously operating since Feb. 8, 2023, the observatory achieves precisions of $2.7\,\mathrm{ppm}$ ($0.58\,\%$) for xCO$_2$ and of $18\,\mathrm{ppb}$ ($0.89\,\%$) for xCH$_4$ respectively, in $5\,\mathrm{min}$ measurements, which is comparable to the pilot study by Griffith et al. (2018), but less precise



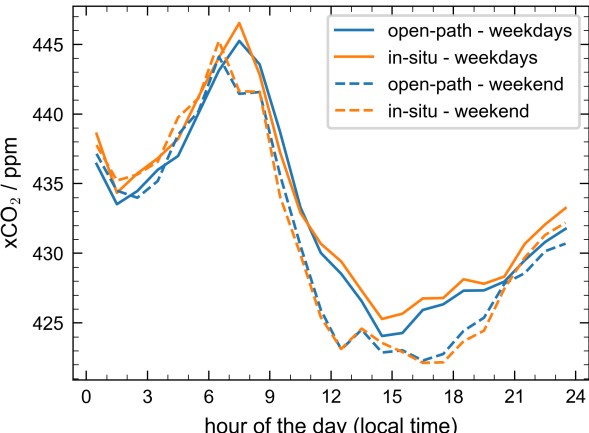

**Figure 11.** Diurnal cycle of $xCO_2$ for weekdays (solid lines) and weekends (dashed lines) respectively. Open-path (blue) and in-situ (orange) are in good accordance at weekends. On weekdays, the in-situ instrument measures systematically 1 to 2 ppm greater values than the open-path instrument.

than the beam splitter coupled setup by Deutscher et al. (2021). Over the whole period, we achieved a data coverage of 78 %
and even up to 92 % for months, were the FTS instrument could be fully dedicated to the open-path measurements. We found
that it is important to consider spectroscopic line-mixing effects of $CO_2$ in the 2 μm band to retrieve accurate path-average
temperatures and that calculating the dry air mole fractions $xCO_2$ and $xCH_4$ using co-measured $O_2$ CDs partially compensates
for systematic errors.

For the period from Feb. 8 to Jun. 26, 2023, we compared the open-path $xCO_2$ and $xCH_4$ abundances to in-situ data measured
at one end of the open-path arrangement, at the IUP. First, we found an overall calibration factor for the open-path data by
scaling to the in-situ data collected under background conditions. After calibration, both records are in good agreement, both
showing a similar interannual trend, as well as diurnal cycles. Then, we focused on showcasing differences between the in-
situ and open-path records that most likely stem from the two methods being representative for different horizontal averaging
scales. We found differences of up to 20 ppm in $xCO_2$ under south-easterly wind directions which suggests that the in-situ
data are more sensitive to localized source patterns than the open-path data. For $xCH_4$, we detected a weak correlation of
the differences with easterly winds which remains unexplained so far. Examining the $xCO_2$ diurnal cycles on weekends and
weekdays, both methods, open-path and in-situ, show a clear signal from traffic emissions during the morning rush hour
but the in-situ instrument reports slightly, but systematically greater $xCO_2$ mole fractions during weekdays. These are all
indications that local emission and transport processes alter the difference between in-situ and open path, which questions
the representativeness of in-situ data. Thus, we recommend evaluating our measurements against atmospheric models that
operate at various horizontal resolutions. This can provide insights on how representative the in-situ and open-path data are on
grid-scales inherent to the models used for quantifying and verifying urban emissions.



Building on the basic observatory setup discussed here and on the versatility of the IFS 125HR FTS, we will aim at refining and expanding the technique in the future. Measuring species such as CO and $NO_2$ together with $CO_2$ might help attribute xCO$_2$ patterns to specific combustion sources. CO observations are currently hindered by the weak transmittance of our cube-corner reflectors in the 2.3 µm range. Thus, replacing the cube-corners will have priority. $NO_2$ has absorption structures in the visible spectral range (0.40 to 0.48 µm) which is within the range of the silicon detector of our FTS. Previous studies have employed DOAS spectrometers to measure $NO_2$ with a similar open-path arrangement as used here. We will also examine whether operating the FTS at higher spectral resolution improves the performance, for example by better separating the target spectral signatures from interfering absorption structures or by reducing systematic errors e.g. due to ILS uncertainties. Open-path measurements might further prove an interesting tool to check and validate spectral line parameters, by providing long absorption paths in a real, but less complex atmosphere than direct sun occultation measurements, which are typically used for these validations. Such checks might include relative and absolute intensity of different absorption bands, but also temperature dependencies of line strengths and differences in pressure broadening by dry and wet air. Especially when analyzing such line broadening effects, high resolution measurements, which fully resolve the spectral line, should provide a clear advantage over lower resolution measurements. Long, continuous time series by our observatory provide a solid data base for such validations. When it comes to improving on the specificity for local emission attribution, we will consider expanding the observatory with additional absorption paths across the city such that the combination of paths yields some primitive horizontal mapping information.

*Data availability.* The data are available from the corresponding author upon request.

*Author contributions.* TDS developed the open-path setup and carried out the formal data analysis. JK and SS supported the development with their experience in open-path measurements. RK supported the development in the lab. BAL supported the data analysis. MS provided the in-situ measurements. SNV advised on the data analysis. FH supported the instrument developed, especially with his expertise in FTIR spectroscopy. DWTG supported the development with the experience and data from his pilot study. TDS and AB wrote the manuscript and all authors commented on the draft. AB conceptualized the project.

*Competing interests.* Some authors are members of the editorial board of AMT. The peer-review process was guided by an independent editor, and the authors have also no other competing interests to declare.

*Acknowledgements.* Many thanks to Samuel Hammer, Susanne Preunkert and Angelika Gassama for keeping the IUP weather station running reliably. Special thanks to Lukas Pilz for his relentless support and push for better software standards and practices, from which the quality of



325 the software within this project profited a lot. The research presented here has been funded by the Deutsche Forschungsgemeinschaft (DFG, German Research Foundation), project number 414273072.



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
