# Peer review of "An open-path observatory for greenhouse gases based on near-infrared Fourier transform spectroscopy"

_Atmospheric Measurement Techniques, 2023_

## Author Response (AR1)

**RC1: 'Comment on amt-2023-185', Anonymous Referee #1, 11 Sep 2023**

https://doi.org/10.5194/amt-2023-185-RC1

**This manuscript is well written and describes the full details of the instrumental setup, as well as spectroscopy measurement and retrieval analysis. It explains clearly the steps of the quantitative calculation and uncertainties. It also states clearly that some instrumental work in this study was built up from previous work and the setup in this study is a more portable application. This study also includes the comparison of the open-path measurement and in situ point measurement over a period of months. It will serve as a very useful reference for open-path FTS measurement of trace gases in regions that have urban other emission sources. Therefore, I recommend publishing this work after addressing my minor comments below.**

We thank the reviewer for the recommendation and appreciation of our work and for the helpful comments. We addressed them point by point below.

**Minor comments:**

**Line 238-248 and Figure 9 are very interesting: (1) How many nights did you find showing similar big differences as March-17 and March-18? Was there any special events? What is the height is the roof top? Were other nights similar or different? Some nights may have higher boundary layer height than other nights. Any daytime observation when the wind was from 135 degrees? (2) Your hypothesis could make sense and I agree that you would need high-resolution models to investigate this further. I wonder if the air mass trajectory models with high-resolution meteorological data would help get an initial estimate of your hypothesis. Did you try any?**

Concerning (1): Manually going through the dataset we identified around 10 nights similar to the ones on March-17 and March-18. All of them show the predominant and steady 135° wind direction. We find only one night with this wind direction which does not show the strong difference. Nights (and days) where the wind does not steadily blow from the 135° direction do not show the difference. We cannot identify any special events linking these nights, except that nearly all of them were in February or March. There are no daytime observations showing a steady wind from the 135° direction since this is predominantly a nighttime wind situation, due to local wind patterns. These local wind patterns are driven by the low mountain ranges in the east, the flats in the west and the river valley.

The height of the instrument roof is around 30 m (depending on the exact location). The open-path telescope is located at 33 m above ground, while the air inlet for the in-situ system is about 30 m above ground, at roughly 20 m horizontal distance from each other. We added information on the height profile along the absorption path to the manuscript.

Concerning (2): We are in exchange on this topic with our local modelling group [e.g. Vardag and Maiwald, GMDD, 2023]. The local mountain-valley topography in the east leads to thermal winds, which need to be simulated on high resolution. However, capturing such effects reliably is very challenging and requires a thorough validation. At the moment, the simulations are not as mature yet as to allow for any robust conclusions. However, the work is ongoing and will benefit from our measurements.

Vardag, S. N. and Maiwald, R.: Optimising Urban Measurement Networks for $CO_2$ Flux Estimation: A High-Resolution Observing System Simulation Experiment using GRAMM/GRAL, Geosci. Model Dev. Discuss. [preprint], https://doi.org/10.5194/gmd-2023-192, in review, 2023

**Figure 10: what is the red solid line? The orientation of the open-path? You need to explain it in the caption.**

We thank you for the remark – this was an oversight by us. You are correct, it is the orientation of the open-path system. We added this to the caption.

**Line 262-264: " … with average differences of less than 1 ppm during the day and slightly larger differences during night." Did you do any test to verify that the difference during night is statistically significant larger than the difference during the day? One issue here is the number of days for weekends + holidays is much less than the number of days for weekdays.**

The slightly higher differences during night than during the day (for weekends) might not be significant and we did not consider it important enough to investigate this further. We just mentioned it for completeness when describing the plot. We agree that the lower number of days for the weekends is an issue which we addressed also in our reply to the comment on Figure 11. To address the statistical uncertainty, we added confidence bands to Fig. 11.

**Line 264-266: how far is the closest major road to your point CRDS measurement?**

The one major road with heavy commuter traffic is roughly 120 m east (nearly in viewing direction of the open-path system) of our Institute, where also the CRDS is situated. We added this information to the caption of Fig. 1.

**Figure 11: the morning peak of xCO2 on the weekends is 1 hour earlier than that on the weekdays. Interesting. Any explanation?**

We noticed this too and do not have a definite explanation. We are uncertain on how a possible sampling bias might play into this, since there are not too many weekend mornings. This might play a role in conjunction with the switch to daylight saving time within the dataset. This time zone switch is of course not relevant for meteorological effects like the boundary layer height, but it affects the human contributions like rush hour traffic. Hence, there is no perfect choice for the time axis (local time or UTC), since either human or natural contributions will be shifted at some point in time and influence the result, also depending on the sampling.

That being said, we could come up with the following speculative explanation: Looking at Fig. 11, measurements on weekends and weekdays mostly agree until 6 a.m. From there on they differ for the next hours by about 5 ppm. This would fit the explanation that the difference is mostly due to higher traffic on weekdays. The impression of the earlier peak might be the result of the tendency of the concentrations to level off between 6 and 8 a.m. together with the subsequent sharp drop. An offset of 5 ppm starting within the plateau would create an earlier drop, creating the illusion of an earlier peak.

For better interpretation of the Fig. 11 we added confidence bands in the plot, which provide the statistical error.

**Line 299-303. Nice. I had similar thoughts before reading this page.**

**RC2**: 'Comment on amt-2023-185', Anonymous Referee #2, 11 Oct 2023
https://doi.org/10.5194/amt-2023-185-RC2

**The manuscript describes an open-path FT spectrometer installation in Heidelberg, Germany with the goal to achieve km-scale measurements of GHG concentrations that could be leveraged to quantify and/or validate urban emissions. The paper provides a clear description of the instrument's progression from previous installations as well as the spectral analysis approach. This work lays the foundation for impactful studies long-term, and I offer some comments below for the authors to consider moving forward.**

We thank the reviewer for the positive assessment of our work and the helpful comments. Please find our point by point reply below.

**Specific Comments:**

**Lines 42 – 50: The authors provide a nice concise description of path-integrated optical spectroscopy methods. Comb-based solutions are highlighted in the context of dual-comb field deployments, but I am curious if the authors considered the impact or potential of coupling a comb source to their FTS system. Spectral bandwidth was mentioned as a potential limiting factor, but recent publications show the thousands-of-wavenumbers metric, highlighted here as a benefit of FTS, is achievable with comb sources. Use of a comb source could address the challenges mentioned with traditional FTS lamp sources.**

We agree. A broadband, bright, and well collimated light source is a major factor for achievable SNR of a path-integrated spectroscopic system. FTS is typically performed using thermal sources – which can only increase their brightness by increasing temperature. This is often not possible or would at least significantly reduce the lifetime of the source, like with halogen lamps. Laser based systems, which can overcome the bandwidth limitations could potentially improve on this. We would like to add a few technical considerations to this idea:

1. Thermal sources also provide the benefit of being relatively quiet, with rather slow drifts in their spectral output. This is important for FTS, since source noise (especially in the kHz region) substantially increases noise in the spectrum or even generates false spectral signals. We tried a laser driven light source as a higher temperature thermal emitter in the past and had too much kHz noise injected, most likely from the laser electronics.
2. Depending on the comb spacing in frequency space, the comb might not provide a "flat" background for high resolution FTS measurements and would generate a type of beating signal with the sampling spacing of the FTS. This would significantly complicate the interpretation and evaluation of the resulting spectra.

We would like to stress, that we currently do not have the technical competencies concerning frequency combs to answers these questions. We consider it an interesting idea and, as highlighted in the manuscript, we are aware of the developments in dual-comb spectroscopy (DCS) and we would be delighted to contribute to side-by-side comparisons of FTS and DCS installations.

**Section 2.6: While comparing path-integrated data to an in situ measurement is often useful for a sanity check, I'm reluctant of "calibrating" the retrievals to a single point measurement. I can view this as a potential bias-correction, but truthfully I would expect other sources of uncertainty (or simple variability in a km-scale field measurement) to dominate over the uncertainties in**

**reference databases which is used here as justification for the correction. I encourage the authors to show a time series overlay of data pre- and post- correction. If there was a way to send the open-path beam into a long-path gas cell filled with the reference material (perhaps possible since the light is fiber coupled?) this would be a more formal way to evaluate bias.**

We agree that "bias correction" would be a better term than "calibrating", also to differentiate it from the high precision calibration that in-situ systems typically perform using calibrated gas mixtures. We changed this in the manuscript.

The scale of the correction factors is actually realistic to be explained by wrong line strengths in the spectroscopic databases. The cited work of Birk et al. (2021) found differences of 2% for $CO_2$ bands just within the 1.6 µm window. We see no reason to assume that the total accuracy of the bands is higher than that. We of course agree with the assessment that variability within a km-scale field measurement can locally dominate over these differences. This makes it challenging to accurately perform these types of bias correction. The derivation of scaling factors from in-situ data and disentangling them from real variability is a general challenge for spectroscopic remote sensing experiments and even well established and longtime operation networks like TCCON face it.

We would like to stress that, in our case, this bias correction is just a global scaling with a constant factor, which was estimated once. A time series of the bias-corrected data would only be shifted by about 8 ppm for $CO_2$ and 20 ppb for $CH_4$ compared to the raw data.

The idea of having a calibration gas cell is intriguing, but would require a multi pass cell with a km-scale absorption path. Something like this would be meters in all dimensions (the light is not as well collimated like a laser after all), a major investment, and is, to our knowledge, only available in a few laboratories to actually perform measurements for spectroscopic databases. It would be further necessary to have precise control over the amount of gas within the absorption cell (per mille scale). To our knowledge this is possible in principle, but it requires a significant amount of effort and attention to detail. Hence, we think that such a gas cell for bias correction is not practical.

**Additional questions that I encourage the authors to consider in this section: When viewing Figure 5, is this data representative of the full campaign of measurements? How often is the in situ measurement system calibrated against reference gases and is there a noticeable drift over time? Is there a particular timescale that is most appropriate for this bias-type correction to be performed? It would be interesting to see if this changes with season or under extreme conditions.**

The data in Figure 5 is taken from a period over 6 weeks in the middle of the campaign. When running the same evaluation on each calendar month separately, we get overall similar results. The deviations in the calibration factor would translate to an uncertainty in the bias correction of below 1 ppm for $CO_2$ and 10 ppb for $CH_4$. Both are well below the 5 minute measurement precision.

The in-situ system is calibrated every 5 hours against reference gases. These frequent calibrations are performed to account for drifts in $^{13}C$ isotope analysis. Mole fraction data of the in-situ system experiences a drift (which is compensated by the calibration) and which is minor within the scope of the comparison (for example 4 ppb methane over 5 years). All of this is described in more detail in the references to the analyzer given in the manuscript (Hoheisel et al., 2019 and Hoheisel, 2021).

We performed this bias correction once in the middle of the timeline to remove the general offset. We do not observe any major drifts yet. We believe that a bias correction at multiple points would run the risk of actually overcompensating for real time variability. We completely agree that changes

with season or extreme conditions will be interesting, also for example for the validation of spectral data. However, we do not have sufficient data to really disentangle many of the meteorological parameters, since they are highly correlated to each other in a dataset which just spans a couple of macro weather situations. The paper here is supposed to lay out the methodologies which, in the future, will be applied to longer timeseries with better potential to disentangle these confounding factors.

**It would also be worth confirming if the Picarro system is using the same version of HITRAN as is being leveraged for open-path fitting. Also, if there is an additional Picarro unit at the other end of the open-path, are the results consistent?**

We do not know which type of cross-section information Picarro uses in their analyzers, since these are proprietary systems concerning hardware and software. High quality in-situ systems correct for long term drifts and biases anyways to achieve their high precision and accuracy goals. Hence, they are regularly calibrated against known gas mixtures. The calibration is often carried outside the manufacturers software, as is the case with the system used within this work.

There is no additional in-situ analyzer at the other end of the open-path although, generally, this is a great idea. At this point, we do not have access to another instrument that we could use.

**Line 253 – Can the authors provide a reference detailing CH4 sources in the area?**

At the moment, there is no peer-reviewed study published which focuses on the composition of CH4 sources within the area, but we added a reference to a Master thesis (Wietzel, 2021) in the manuscript. Most of the information therein is from local and state authorities and the EDGAR inventory. Hoheisel et al. (2019) further discusses regional sources of CH4.

Wietzel, J. B.: Estimation of Methane Emissions in the Urban Area of Heidelberg by Mobile Measurements, Master's thesis, Heidelberg University, 2021.

**Technical Corrections:**

**Line 207 – "precise" should be "precision"**

We corrected this in the manuscript.

**Line 238 – "extend" should be "extent"**

We corrected this in the manuscript.

RC3: 'Comment on amt-2023-185', Anonymous Referee #3, 11 Oct 2023

https://doi.org/10.5194/amt-2023-185-RC3

**This paper demonstrates open-path Fourier transform spectroscopy over a 1.55 km long path using a high resolution spectrometer (Bruker 125HR). The system operated for about 5 months, and the data were compared to an in situ sensor for CH4 and CO2. After an offset correction, the measurements agree well on average with some differences observed or specific wind directions and plumes. Overall, the paper was well written and provides some useful technical details. I do have a few questions:**

We thank the reviewer for the appreciation of our work and the helpful questions and comments. Please find our point by point reply below.

**Is there any active pointing feedback during the measurements?**

No, there is not. Currently the telescope is in a fixed installation. Note that due to beam divergence, the cross section of the light beam on the far side of the absorption path is bigger than the reflector array. Consequently, the array is always fully illuminated even if there are turbulent fluctuations.

**Could you show an Allan-Werle deviation plot for some period or periods of concentration measurements, especially for the different measurement configurations?**

[Figure]

*Figure AC1: Allan deviations for CO2 (left) and CH4 (right), each for open-path (blue) and in-situ (orange). Dashed lines give the purely statistical extrapolation from the first data point.*

The measurement configurations which average over 5 minutes are already close to the point where natural variability is a dominant factor on the signal. Hence, an Allan deviation plot is difficult to interpret. But for the 1 minute measurement mode, such a plot is instructive:
Fig. AC1 shows Allan deviations for CO2 and CH4. It is clearly visible, that the lowest averaging time (1 minute) is already too coarse for the precision of the in-situ analyzer. The open-path instrument reaches its minimum Allan deviation somewhere between 10 and 30 minutes and increases for longer averaging times. For these time scales, in-situ and open-path instrument are both limited by the natural variability and follow the same pattern.

**The residuals seem to be dominated by low to medium frequency structure, not white noise. What's the limiting noise source? I think nonlinearities can impact the baseline, do you perform any nonlinearity corrections?**

[Figure]

*Figure AC2: Histogram of the noise band (left) of a single spectrum with a fitted Gaussian. Errors are statistical. Histogram of a single residual in the 1.6 µm band in units of the estimated white noise, after first order correction for systematic residuals. Errors are statistical.*

The noise of single spectra is definitely dominated by white noise, following a Gaussian distribution (Fig. AC2). For some bands (especially in the 2 µm band) we see that the systematic residual is large enough to have a significant impact. If we subtract the systematic residual in a first order approximation from a single residual, we get a Gaussian distribution with a width nearly matching our expectation from our white noise estimation for all spectral bands.

Nonlinearities of the detector would definitely impact the baseline. However, this is typically not an issue with InGaAs photodiodes, especially with such low light conditions and a photocurrent in the region of µA. Hence, we do not perform a nonlinearity correction.

**The description of the "background" determination was a bit confusing and seems like a somewhat arbitrary definition of the background. What does a scatter plot of concentrations look like? Do you get similar results? Or, have you tried to use a baseline fitting method, for example from the pybaselines package?**

The definition of the "background" is somewhat arbitrary since it is not possible to have a measurement without any enhancement. Our method described in the manuscript tries to identify measurements without significant local enhancement. We judged this to be a better method than just using metrics like a minimum wind speed to identify negligibly enhanced measurements.

[Figure]

*Figure AC3: Scatter plot of open-path measurements before bias correction vs. in-situ measurements. The orange dotted line is the identity. The black dashed line is a fit of a scaling factor m.*

Fig. AC3 shows a scatter plot of open-path measurements before bias correction plotted against in-situ concentrations. For both CO2 and CH4 the result from a simple fit of a scaling factor produces similar results to our bias estimation.

We did not try any approaches with baseline fitting, since the real enhancements/differences can last for tens of hours and are thus not differentiable to a background "baseline" in a time series.

**The methane open path histogram seems to be fairly symmetric (i.e., Gaussian). However, the FWHM does not agree with the standard deviation determined by the point-by-point difference. Given the standard deviation of 18 ppb, I would expect a noise-limited FWHM = 2.355*\sigma = 42.4 ppb. However, the width from Fig 5 is more like 120 ppb. Do you know why?**

This is a result of the actual variability of the signal which is mixed into this plot. We think, that for this sanity check it is more appropriate to look at Fig. 8: The methane histogram for the differences is also somewhat Gaussian. Assuming the in-situ data as something like the "ground truth" for this purpose and in knowledge of the fact that the in-situ measurement precision is negligible when compared with the open-path system, we can see a FWHM of around 60 ppb, which is a lot closer to the estimation purely explained by statistical variability and instrument precision. We attribute the remaining 20 ppb to the fact, that the in-situ data is of course not the "ground truth" for the open-path system.

**It wasn't clear to me for which data the calibration factors (relative to in situ) had been applied. Were they applied to the all data shown in Fig 7 and beyond?**

Yes, they were applied to Fig 7 and beyond. We refined this in the manuscript to make it more clear.

**For the ILS correction terms, how stable do you think these terms are? Did you repeat the ILS determination on other days?**

There are no corrections with respect to the ILS for individual spectra. We evaluate the ILS once (for each resolution) and then keep it constant for the whole dataset when running our spectral fits to retrieve trace gas information. We evaluated the ILS a few times within the time series and could only find minor changes – nothing significant within the sensitivity of our ILS retrieval. Hence, we

prefer to evaluate the whole time series with a constant ILS to not inject drifts or jumps into the data.

**Another possibility for the differences at 135 degrees would be a local source near the in situ sensor at the direction. Do you see this difference during day and nighttime? Are there any potential sources on nearby buildings?**

We totally agree, but could not identify a suitable source. There is no building in this direction. 120m east of the institute is a major road with heavy commuter traffic traversing in direct north-south direction, which would be the most obvious emission source in the direct vicinity. But there are several details that do not fit this explanation: We see the signal most dominantly at night, when we have a strong boundary layer and a corresponding build up. While there is also some traffic at night, it does not necessarily fit the magnitude of the buildup. Also, we do not see the signal differences for winds from the east or north-east, which would also bring air from the road to the in-situ analyzer. The only explanation would be a more complex interaction with the surrounding buildings, why these metrological conditions then do not produce this signal. An explanation like this would again need fine resolution modelling to be convincing.

**Have you used other O2 lineshape models, i.e., those from:**

**H. Fleurbaey, Z. D. Reed, E. M. Adkins, D. A. Long, and J. T. Hodges, "High accuracy spectroscopic parameters of the 1.27 µm band of O2 measured with comb-referenced, cavity ring-down spectroscopy," J. Quant. Spectrosc. Radiat. Transfer 270, 107684 (2021).**

**Other recent open-path measurements have shown a 1% difference between retrievals with the two models:**

**Malarich, N.A., Washburn, B.R., Cossel, K.C., Mead, G.J., Giorgetta, F.R., Herman, D.I., Newbury, N.R., Coddington, I., 2023. Validation of open-path dual-comb spectroscopy against an O2 background. Opt. Express 31, 5042–5055. https://doi.org/10.1364/OE.480301**

We only used line shape parameters and models as described within the paper and did not test more complicated parameter sets or models like the referenced β-qSDNGP profile. We agree, that this is an interesting idea and, as mentioned in the last paragraph of our manuscript, we plan to investigate in the future how we can contribute in the validation process of spectroscopic parameters like line-by-line data and line-shape models. We added a reference to Malarich et al. (2023) as an additional example for uncertainties and offsets produced by spectroscopic databases.

**In addition, I have a few minor comments:**

**In a couple of spots, "extend" was used instead of "extent"**

We corrected this in the manuscript.

**"disposes of" probably should be "consists of" or something similar?**

We corrected this in the manuscript.

**It would be nice to have an elevation profile along the measurement path**

Thank you for this comment, we missed to give this information. The elevation profile is rather unexciting: The terrain is mostly flat (variations are contained within 111 m and 116 m above sea level). In the last 200 meters close to the old physics institute we have a quasi linear increase to 137 m above sea level. We added this information in the manuscript.

**Maybe I missed it, but what is the red line on figure 10?**

We thank you for the remark – this was an oversight by us. It is the viewing direction of the open-path system. We added this to the caption.